# The Presence of Blood–Brain Barrier Modulates the Response to Magnesium Salts in Human Brain Organoids

**DOI:** 10.3390/ijms23095133

**Published:** 2022-05-04

**Authors:** Alessandra Cazzaniga, Giorgia Fedele, Sara Castiglioni, Jeanette A. Maier

**Affiliations:** 1Department of Biomedical and Clinical Sciences, Università di Milano, 20157 Milano, Italy; giorgia.fedele@unimi.it (G.F.); sara.castiglioni@unimi.it (S.C.); jeanette.maier@unimi.it (J.A.M.); 2Interdisciplinary Centre for Nanostructured Materials and Interfaces (CIMaINa), Università di Milano, 20133 Milano, Italy

**Keywords:** induced pluripotent stem cells, organoids, GABA receptors, NMDA receptor, BDNF, magnesium

## Abstract

Magnesium (Mg) is fundamental in the brain, where it regulates metabolism and neurotransmission and protects against neuroinflammation. To obtain insights into the molecular basis of Mg action in the brain, we investigated the effects of Mg in human brain organoids, a revolutionary 3D model to study neurobiology and neuropathology. In particular, brain organoids derived from human induced pluripotent stem cells were cultured in the presence or in the absence of an in vitro-generated blood–brain barrier (BBB), and then exposed to 1 or 5 mM concentrations of inorganic and organic Mg salts (Mg sulphate (MgSO_4_); Mg pidolate (MgPid)). We evaluated the modulation of NMDA and GABAergic receptors, and BDNF. Our data suggest that the presence of the BBB is essential for Mg to exert its effects on brain organoids, and that 5 mM of MgPid is more effective than MgSO_4_ in increasing the levels of GABA receptors and BDNF, and decreasing those of NMDA receptor. These results might illuminate novel pathways explaining the neuroprotective role of Mg.

## 1. Introduction

The possibility to culture human pluripotent stem cells, which are capable of self-renewal and self-organization, paved the way for the development of organoids; these are 3D multicellular clusters which provide a unique opportunity to model human organogenesis, and resemble the structure of human organs [1]. In particular, brain organoids recapitulate the structural and organizational complexity of the human brain such as the cortical layering and the interconnectivity between different regions [2]. Notably, they have been used to study brain development, to investigate the pathogenesis of neurodevelopmental disorders and neurodegenerative diseases, and for drug screenings [3]. 

Here, we investigate the effects of two Magnesium (Mg) salts—Mg sulphate (MgSO_4_) and Mg pidolate (MgPid)—in brain organoids generated from human induced pluripotent stem cells (iPSCs). The rationale for using these two salts lies in the fact that the first is an inorganic salt, and the second is organic. These two salts have been previously tested on an in vitro model of the murine blood–brain barrier (BBB), and MgPid was shown to cross the BBB more efficiently than MgSO_4_ [4].

Mg is a vital metabolite with essential roles in energetic metabolism, calcium homeostasis, enzymatic activity and transcellular transport [5]. In the brain, Mg excites γ-aminobutyric acid receptors (GABA-Rs) and inhibits N-methyl-D-aspartate receptor (NMDA-R) [6,7,8]. It stabilizes the BBB [4], modulates the hypothalamic pituitary adrenal axis [9], and is involved in regulating the transcription of genes implicated in neurotransmission [10]. Mg has also a role in the regulation of neuropeptide release and in counteracting oxidative stress, thus contributing to the maintenance of healthy neurological function [11,12]. Therefore, it is not surprising that an imbalance of Mg homeostasis has been associated with neurological diseases and psychiatric disorders [6]. Indeed, as mentioned above, Mg antagonizes NMDA-R, whose overactivation is linked to neuronal injury in several acute and chronic disorders, including Alzheimer’s and Huntington’s diseases, stroke, and traumatic brain injury [13]. While more research is needed on Mg as an adjunct treatment in Alzheimer’s and Parkinson’s diseases, emerging data support the protective effect of Mg in stroke and traumatic brain injury [6], and the administration of Mg is recommended to prevent maternal seizures associated with preeclampsia [14]. Mg also ameliorates anxiety and stress by activating the GABAergic pathway [9,15]. γ-aminobutyric acid (GABA) is the major inhibitory neurotransmitter in the central nervous system and acts through the ionotropic GABA type A (GABA_A_) and metabotropic GABA type B (GABA_B_) receptors. Once GABA binds its receptors, they become permeable to chloride influx, which suppresses neuronal activity in the brain [16]. Dysfunction of these GABA-Rs is associated with anxiety, epilepsy, and neurodevelopmental disorders, including autism [15]. Of importance, the deficiency of GABA_A_ receptors results in the overactivity of NMDA-R and, therefore, excitotoxicity [17].

Glutamate via NMDA-R and GABA via GABA_A_-R have been shown to upregulate brain-derived neurotrophic factor (BDNF) in hippocampal and hypothalamic neurons [18]. BDNF is a neurotrophin involved in the development and maintenance of the nervous system [19] and is synthesized not only by neurons, but also by cerebral endothelium. Endothelial-derived BDNF protects neurons from oxidative stress, hypoxia and amyloid toxicity [20]. Accordingly, unhealthy cerebral endothelium is associated with neuronal dysfunction [21].

We anticipate that using brain organoids might broaden present knowledge about the effects of Mg in the human brain. We hypothesize that Mg salts might modulate the levels of NMDA-R, GABA-Rs and BDNF in organoids generated by iPSCs. We also wonder whether the presence of an in vitro model of the BBB might influence the response to Mg salts. Indeed, one of the main function of the BBB is to regulate the transport of molecules into and out of the central nervous system to preserve the proper chemical composition of the neuronal milieu necessary for neuronal function [22]. Consistently, the loss of BBB integrity or changes in its permeability are associated with several neuropathological disorders, and the treatments aimed at stabilizing the BBB are scarce [23,24]

## 2. Results

### 2.1. Generation of Brain Organoids in the Presence or Absence of the BBB

We developed a co-culture model of the BBB and brain organoids. The timeline of the experiment is shown in Figure 1a. Based on previous studies [4,25], iPSCs were cultured in appropriate media to form an embryoid body (EB), which was then induced to neuroectodermal fate. The neuroectodermal progenitors self-organize into 3D structures. Then, the neuroectodermal cells generate the neural stem and progenitor cells that proliferate and differentiate into neurons and glia. Once they start to differentiate, the tissues become more complex, as indicated by the budding outgrowth (Figure 1a). After 35 days, the organoids are mature (Figure 1a). Immunofluorescence reveals a complex morphology with heterogeneous regions containing neurons (GABA_B_-R, green) and astrocytes (S100β, red). As described [25], the neuroepithelial cells organize into 2D polarized, rosette-like formations (white arrows) (Figure 1a). At this time point, brain organoids are placed under the BBB for 4 additional days in the presence of 1 or 5 mM MgSO_4_ or MgPid (Figure 1b).

### 2.2. NMDA-R in Brain Organoids in the Presence or Absence of the BBB Exposed to Different Concentrations of MgSO_4_ or MgPid

NMDA-R is a subtype of glutamate receptor which is inhibited by Mg [8]. We focused on the ubiquitously expressed and obligatory subunit type 1 of the receptor. We measured the transcript using real-time PCR and the total protein amount using Western blot in organoids co-cultured with the in vitro-generated BBB (BBB), or in their controls, in the absence of the BBB (CTR). Only when the organoids were maintained under the BBB 5 mM MgPid significantly reduce both the *NMDA-R* transcript (Figure 2a) and the total amount of the protein (Figure 2b,c). NMDA-R downregulation was also observed at the protein level in response to 5 mM MgSO_4_, but it does not reach statistical significance (Figure 2b,c). Notably, pidolic acid did not exert any effect on *NMDA-R* expression (Appendix A). 

### 2.3. GABA-Rs in Brain Organoids in the Presence or Absence of the BBB Exposed to Different Concentrations of MgSO_4_ or MgPid

We evaluated, using real-time PCR and Western blot, the modulation of GABA-Rs in the presence or absence of the BBB after exposure to 1 or 5 mM of MgSO_4_ and MgPid for 4 days. At the transcriptional and protein levels, GABA_A_-R is only significantly overexpressed in the presence of the BBB upon exposure to 5 mM of MgPid (Figure 3a,c,e). On the contrary, GABA_B_-R is significantly increased with both 5 mM of MgSO_4_ and MgPid in the presence of the BBB (Figure 3b,d,e). No differences in GABA-Rs were detected in the absence of the BBB. Pidolic acid had no effect on *GABA-Rs* expression (Appendix A).

### 2.4. Mg Transport across the BBB—Brain Organoids Exposed to Different Concentrations of MgSO_4_ and MgPid

Since differences in NMDA-R and GABA-R expression have only been detected in brain organoids after exposure to 5 mM Mg in the presence of the BBB, we analyzed how much Mg crosses the BBB by measuring total Mg with the fluorescent probe DCHQ5. As a control, we maintained the organoids in the presence of a transwell insert without the BBB. After 4 days, the incubation with 5 mM Mg salts increased the passage of Mg through the BBB (Figure 4a), and MgPid was slightly more efficient than MgSO_4_. The insert alone (CTR) did not prevent the passage of the two salts. We also measured total Mg in the organoids and found it increased after incubation with 5 mM Mg salts (Figure 4b). MgPid in the amount of 5 mM was more efficient than 5 mM MgSO_4_ in increasing intra-organoid Mg in the presence of the BBB.

### 2.5. BDNF Levels in Brain Organoids in the Presence or Absence of the BBB Exposed to Different Concentrations of MgSO_4_ and MgPid 

Because the neurotrophic factor BDNF is critical for neuron survival, synaptogenesis and neurotransmission, we wondered if different concentrations of Mg impact BDNF levels in our experimental model. Using real-time PCR, we only observed an overexpression of *BDNF* in brain organoids under the BBB exposed to 5 mM MgPid for 4 days (Figure 5a). This result was reflected at the protein levels, as detected by ELISA (Figure 5b). Without the BBB, the levels of BDNF are low and not modulated by Mg salts. We also measured BDNF in the lower chamber by ELISA. In the presence of the BBB, more BDNF was detectable in the medium. Interestingly, 5 mM MgPid increased the released BDNF (Figure 5c). Again, the exposure with pidolic acid had no effect (Appendix A).

## 3. Discussion

Beyond its critical role in metabolism, Mg is crucial in the nervous system because of its neuroprotective action against excitotoxicity by acting as an antagonist of NMDA-R. Mg also potentiates GABA signaling, controls nerve conduction and neuromuscular transmission, prevents oxidative stress, and downregulates inflammatory mediators [6]. Clinical studies indicate that patients with mood disorders show deranged Mg homeostasis [26]. Accordingly, administering Mg has a beneficial effect on anxiety [9]. A recent clinical trial concluded that Mg supplementation provides significant clinical benefits in stressed healthy adults [27]. Mg seems to be beneficial in depression [28] and in relieving headache [29]. Therefore, there is an increasing interest in Mg as a molecule capable of harmonizing neuronal circuits. To understand how Mg affects the nervous system, animal models have been utilized and have provided valuable insights into the mechanisms responsible for neurological and behavioral disorders under Mg-deficient conditions [30,31]. However, translatability is often limited because of the major differences between the brains of humans and animals [32]. Three-dimensional brain organoids emerged as a revolutionary tool to study the response to various compounds. Therefore, to obtain novel insights into the mechanisms underlying the Mg effect in the brain, we utilized brain organoids exposed to two different Mg salts: the inorganic MgSO_4_ and the organic MgPid, which seem to possess different bioavailability [4,33]. Moreover, because of the crucial role of the BBB in controlling the exchange to and from the brain [34], we compared the effects of the two different concentrations of Mg salts on brain organoids maintained, or not, under an in vitro-generated BBB. This is a rather simple experimental model that allows us to better mimic what happens in vivo. Indeed, significant differences emerged in the response to Mg salts between brain organoids in the presence or absence of the BBB. We began by analyzing the levels of NMDA-R and GABA-Rs. It is known that Mg affects the function of these receptors [7,35], but no data are available on their amounts after exposure to high concentrations of Mg. In brain organoids cultured in the absence of the BBB, no significant changes in the levels of these receptors were observed after 4 days of culture in 1 or 5 mM of Mg salts, thus suggesting that Mg alone is not sufficient to modulate their expression. It is the presence of the BBB that grants the possibility to detect the effects of 5 mM MgSO_4_ or MgPid in modulating these receptors. Indeed, we found that 5 mM of both salts increased the amounts of the metabotropic GABA_B_-R, mainly through transcriptional regulation. GABA_B_-R dysregulation is involved in many neurological and psychiatric disorders, from epilepsy to depression and anxiety, and from drug addiction to sleep disorders [36]. At the moment, GABA_B_-R agonists are under investigation to treat alcohol addiction in humans [37]. It is feasible to propose Mg as an adjuvant in all these conditions. Only 5 mM MgPid upregulated the ionotropic GABA_A_-R. Interestingly, GABA_A_-Rs is the target of benzodiazepine, known to exert its anxiolytic, sedative, hypnotic and anticonvulsant effects by enhancing its function [38]. We anticipate that MgPid and benzodiazepine might have an addictive effect, the first because it augments the amounts of GABA_A_-R, the second because it activates the receptor. On this basis, it will be interesting to investigate whether the response to benzodiazepines might depend on the individual Mg status. The presence of the BBB is also essential in observing the downregulation of NMDA-R by 5 mM MgPid and, to a lesser extent, MgSO_4_. It is well known that Mg blocks NMDA-R activation [8]. Our data indicate that Mg also reduces the total amounts of NMDA-R, an event which might be useful to prevent excitotoxity and, consequently, neuronal death. This is the first demonstration of an effect of Mg on the total amount NMDA-R, generally considered to be rather stable. In general, aberrant or pathological NMDA-R effects occur mainly via abnormal receptor activity, and far less is known about its expression. It is known that NMDA-R is upregulated by ethanol in cortical neurons [39,40], and this contributes to ethanol-induced neurotoxicity. In rats, chronic stress increases the protein levels of NMDA-R in the hippocampus, an event that may induce excitatory neurotoxicity [41]. Interestingly, the deletion of the obligatory subunit 1 of NMDA-R in the ventral hippocampus reduces anxiety [42]. Our data suggest that MgPid might be a useful tool to control NMDA-R, not only by inhibiting its activity, but also by downregulating it.

We measured Mg concentration in the lower chamber and in the brain organoids, and found that it markedly increased with 5 mM salts both in the presence and absence of the BBB. In agreement with previous findings, MgPid is more efficient than MgSO_4_ in crossing the BBB [4]. Moreover, 5 mM MgPid led to higher levels of intra-organoid Mg than 5 mM MgSO_4_. It is likely that MgPid is more effective than MgSO_4_ because of its higher bioavailability, which might facilitate its entry in the cells [4,29]. Therefore, Mg alone does not explain the results reported above and, specifically, the significant differences between the activity of MgPid and MgSO_4_ in brain organoids cultured under the BBB. We propose that, beyond its role as a barrier, the BBB also exerts modulatory functions on the organoids by secreting bioactive molecules.

The neurotrophin BDNF is a crucial player in neurodevelopment, neuroplasticity and neurosurvival [43]. Notably, several authors add BDNF to the medium of the organoids to optimize neural induction [44]. Our results showed that BDNF levels were significantly upregulated in brain organoids under the BBB exposed to 5 mM MgPid. Moreover, 5 mM MgPid increased the release of BDNF. These results are in agreement with the study showing that Mg elevates the expression of BDNF in the amygdala, in the prefrontal cortex and in the hippocampus of rats subjected to olfactory bulbectomy, an in vivo model of depression [45]. Interestingly, treatment with Mg and vitamin D augments serum BDNF levels in patients affected by depression and obesity [46]. Moreover, it has been observed that an increase in Mg level in the brain is associated with the enhancement of BDNF [28]. It is known that the activation of GABA_B_-R triggers the release of BDNF, which then increases the level of GABA_A_-R by activating the JAK/STAT pathway [47]. Moreover, BDNF inhibits the endocytosis of GABA_A_-R through the activation of the protein kinase C pathway, with a consequent increase in the cell surface receptors [48]. Furthermore, an inverse relation exists between GABA_A_-R and NMDA-R [49]. At the moment, we can only speculate that similar events might occur in brain organoids exposed to 5 mM MgPid in the presence of the BBB [28]. More studies are necessary to understand the links between high levels of BDNF and the regulation of the abundance of GABA-Rs and NMDA-R by MgPid.

Low levels of brain BDNF are suspected to be involved in the cognitive impairment associated with stress-related mood disorders [50]. Moreover, brain *BDNF* mRNA levels correlate with Mini-Mental State results in aged patients [51]. Accordingly, BDNF is emerging as a cornerstone in the treatment of psychiatric disorders [43], and MgPid might contribute to alleviating the symptoms by increasing BDNF levels in the brain.

We are aware of several limitations of our study. First, we measured the abundance of the receptor at the RNA and protein levels, but no experiments were performed on their activity. Second, all the experiments were performed on brain organoids after 35 days of culture/growth, while analyzing the brain organoids at different time points might yield interesting insights to better understand the regulation of the expression of these proteins. Moreover, it would be interesting to extend these studies to the various subunits of NMDA-R and GABA-Rs testing also other organic Mg salts. 

Although more basic and translational studies are required, this “pilot” study underscores that (i) the BBB is necessary for Mg salts to exert effects in the brain organoids, and (ii) MgPid performs better that MgSO_4_ in modulating the abundance of NMDA-R, GABA-Rs and BDNF.

## 4. Materials and Methods

### 4.1. Generation of Human Brain Organoids

The human episomal induced pluripotent stem cells (iPSCs) were purchased (Gibco, Thermo Fisher Scientific, Waltham, MA, USA) and cultured in mTesR culture media (Stem Cell Technologies, Vancouver, Canada) following the manufacturers’ instructions. The cells were routinely tested for the expression of stemness markers and used for 4–6 passages. iPSCs were plated in a dish pre-coated for 1 h at room temperature at 37 °C with Matrigel^®^ hESC-Qualified Matrix (Corning, Corning, NY, USA). For the generation of brain organoids, the cells were detached using a gentle cell dissociation reagent (Stem Cell Technologies) once they reached 80% of confluence, as indicated in the Lancaster protocol [25]. Shortly after, to generate embryoid bodies (EBs), 9000 cells/well were plated in an ultra-low-attachment 96-well plate (Corning) and maintained in media containing 4 ng/mL basic fibroblast growth factor (bFGF) (Thermo Fisher Scientific, Waltham, MA, USA), which maintains the pluripotency of iPSCs [52], and 10 µM Rock-inhibitor Y-27632 (Sigma-Aldrich, St. Louis, MO, USA) to increase cell survival [25,53]. After 7 days, EBs reached approximately 500 µm in diameter and were transferred into a neural induction medium for 4 days. At this point, neuroepithelia could be visualized. Then, the neuroectoderm formations were embedded in the center of a droplet of Matrigel^®^ hESC-Qualified Matrix (Corning) and cultured for another 4 days in a differentiation medium without vitamin A. At this stage, the tissues had become more complex, with some budding outgrowth and radial processes. From this point forward, the early brain organoids were maintained on an orbital shaker to permit the exchange of nutrients and oxygen. Finally, they were transferred into a new differentiation medium, added with vitamin A, for the last twenty days in order to obtain mature human brain organoids. To analyze the RNA and protein content, each experimental sample was obtained by pooling three organoids. Each experiment was repeated at least three times.

Immunofluorescence was performed after 35 days of culture. Sections (10 µm) were stained with anti-GABA_B_-R to detect neurons [54], and anti-s100β to visualize astrocytes [55]. Alexa Fluor 488 (green) and 546 (red) (Thermo Fisher Scientific) were used as secondary antibodies. 4′,6-diamidino-2-phenylindole (DAPI) (Sigma Aldrich) was used to stain the nuclei. Finally, the slices were mounted with moviol and images were acquired with a 40X objective in oil using an SP8 confocal microscope (Leica Microsystems, Buffalo Grove, IL, USA).

### 4.2. In Vitro Model of Blood Brain Barrier (BBB) and Brain Organoids

Human brain microvascular endothelial cells (HbMEC) (Innoprot, Bizkaia, Spain) were cultured in endothelial medium (ECM) (Innoprot) supplemented with 5% fetal bovine serum (FBS), 1% endothelial cell growth supplement (ECGS) and a 1% penicillin/streptomycin solution (P/S). Human Astrocytes (HA) (Science Cell, San Diego, CA, USA) were cultured in their medium containing 2% FBS, 1% astrocyte growth supplement and 1% P/S. The BBB in vitro model was obtained by exploiting the transwell system (Corning) with a 0.4 µm pore size. Firstly, the underside and the upper side of the transwell insert were coated with Poly-L-lysine (2 µg/mL) and fibronectin (50 µg/mL), respectively. Thereafter, HAs were seeded (35,000/cm^2^) underneath the transwell insert and HbMECs were seeded on the upper side (60,000/cm^2^) [4].

1 mM is considered the physiological concentration for Mg. To perform the experiments with 1 and 5 mM Mg sulphate (MgSO_4_) or Mg pidolate (MgPid), a custom-made Mg-free minimum essential medium (MEM) was used (Thermo Fisher Scientific). MgSO_4_ or MgPid were added to this medium to reach the above-mentioned final concentrations. Pidolic acid was used as a control (Appendix A). The medium containing 1 or 5 mM Mg was used in the upper chamber of the transwell system, while in the lower chamber, the medium containing the physiological Mg concentration (1 mM MgSO_4_) was used.

### 4.3. Real-Time PCR

Total RNA from three-pooled organoids was extracted using the Purelink RNA Mini Kit (Invitrogen, Thermo Fisher Scientific). Single-stranded cDNA was synthesized from 0.1 µg RNA in a 20 µL final volume using the High-Capacity cDNA Reverse Transcription Kit with RNase inhibitor (Applied Biosystems, Thermo Fisher Scientific) according to the manufacturer’s instructions.

Real-time PCR was performed three times, in triplicate, on the CFX96 Real-Time PCR Detection system (Bio-Rad, Hercules, California, USA), and TaqMan Gene Expression Assays probes were used (Life Technologies, Monza, Italy). The following probes were used: 1. targeting *GRIN1* (*NMDA-R*) (Hs00609557_m1), coding for the ubiquitously expressed and obligatory NMDA type subunit 1; 2. *GABBR1* (*GABA_B_-R*) (Hs00559488_m1), coding for GABA type-B receptor subunit 1; 3. *GABRA2* (*GABA_A_-R*) (Hs00168069_m1), coding for GABA type-A receptor α2; and *brain-derived neurotrophic factor* (*BDNF*) (Hs02718934_s1). The housekeeping gene glyceraldehyde-3-phosphate dehydrogenase (*GAPDH*) (Hs99999905_m1) was used as an internal reference gene. Relative changes in gene expression were analyzed using the 2^−∆∆Ct^ method. The experiment was performed in triplicate at least three times.

### 4.4. Western Blot

Pools of three organoids were lysed in 50 mM Tris–HCl (pH 7.4) containing 150 mM NaCl, 1% NP40, 0.25% sodium deoxycholate, protease inhibitors (10 μg/mL Leupeptin, 10 μg/mL Aprotinin, 1 mM PMSF) and phosphatase inhibitors (1 mM sodium fluoride, 1 mM sodium vanadate, 5 mM sodium phosphate). The concentration of protein in samples was assessed using the Bradford assay (Sigma Aldrich). Lysates (40 μg/lane) were separated on SDS-PAGE and transferred to nitrocellulose sheets using the Trans-Blot Turbo Transfer System following the manufacturer’s instructions (Bio-Rad). Western Blot analysis was performed using antibodies against NMDA type subunit 1 (NMDA-R), the key subunit of the NMDA receptor (Invitrogen) [56], GABA type-B receptor subunit 1 (GABA_B_-R) (Abnova, Taipei, Taiwan), and GABA type-A receptor α2 (GABA_A_-R) (Thermo Fisher Scientific) which are the most abundant subunits expressed in most brain regions [57,58], and β-actin (Santa Cruz Biotechnology, Dallas, TX, USA).

After washing, secondary antibodies labelled with horseradish peroxidase (Amersham Pharmacia Biotech Italia, Cologno Monzese, Italy) were used. The immunoreactive proteins were detected using ClarityTM Western ECL substrate (Bio-Rad), and images were captured using a ChemiDoc MP Imaging System (Bio-Rad). The independent experiment was repeated at least five times. Densitometry was performed using the software ImageJ (National Institute of Health, Bethesda, MD, USA). A representative blot of at least five independent experiments is shown.

### 4.5. ELISA Assay

To determine the protein amount of human BDNF in our samples, the ELISA kit for human BDNF protein (Abcam, Cambridge, UK) was used on brain organoid extracts cultured in the presence or absence of the BBB, and on their respective culture media, according to the manufacturer’s instructions. The ELISA was performed in triplicate and at least three times on 30 μg of brain organoid extracts, or on 50 µL of culture media.

### 4.6. Evaluation of Mg Content

Total Mg levels in the lower chamber and in the lysates of the brain organoids were measured using the fluorescent dye based on diaza-18-crown-6-hydroxyquinoline (DCHQ5) (kindly donated by Prof. S. Iotti, University of Bologna) as described [59]. The lysis of brain organoids was conducted through repeated freezing-and-thawing cycles in phosphate-buffered saline (PBS) (Euroclone, Milan, Italy). Then, the protein content was quantified using the Bradford assay, and 30 µg of each sample was used to determine Mg concentration. Fluorescence intensities were acquired at 510 nm using the Varioskan LUX Multimode Microplate Reader (Thermo Fisher Scientific). Mg concentrations were obtained through interpolation of the samples’ fluorescence with the standard curve, previously performed using known concentrations of MgSO_4_. All the experiments were performed in triplicate at least three times.

### 4.7. Statistical Analysis

Data are reported as medians and interquartile range. The data were analyzed using two-way ANOVA. The *p*-values deriving from multiple comparisons were corrected using the Tukey method. Statistical significance was defined for *p*-value ≤ 0.05 (* *p* ≤ 0.05; ** *p* ≤ 0.01; *** *p* ≤ 0.001).

## Figures and Tables

**Figure 1 ijms-23-05133-f001:**
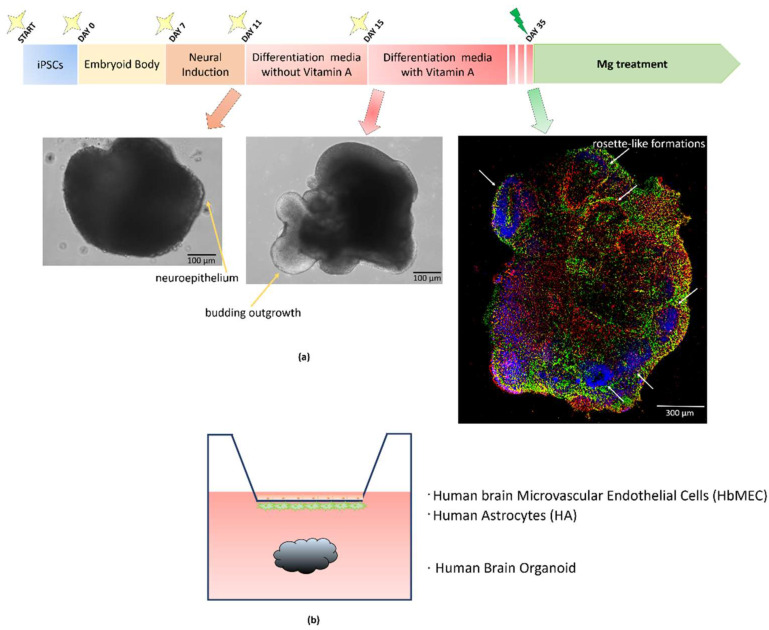
Graphical representation of the experimental model: (**a**) The timeline of human brain organoid generation is shown. Representative brain organoid with neuroectoderm (yellow arrow) and with budding outgrowth (yellow arrow) are shown. Immunofluorescence staining of a brain organoid after 35 days in culture. Antibodies against GABA_B_-R (green) and anti-astroglial calcium-binding protein S100β (red) were utilized. DAPI (blue) labels the nuclei. Rosette-like formations are shown (white arrows); (**b**) An illustrative scheme of the BBB/organoid co-culture is shown.

**Figure 2 ijms-23-05133-f002:**
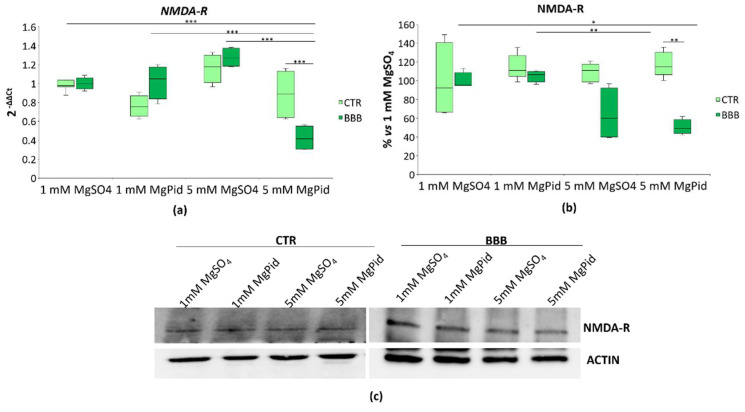
NMDA-R in brain organoids in the presence or absence of the BBB. Human brain organoids were cultured with (BBB) or without (CTR) the BBB, and exposed to 1 or 5 mM MgPid or MgSO_4_ for 4 days. Each experimental sample was obtained by pooling three organoids. (**a**) Box plot shows the real-time PCR performed three times in triplicate on RNA extracted from the brain organoids. Primers were designed on *GRIN1* (*NMDA-R*) sequence. The mRNA expression values were normalized to *GAPDH* mRNA levels; (**b**) Western blot was performed using antibodies against the subunit 1 of the NMDA-R. Actin was used as a control of loading. Box plot shows the quantification of five independent Western blots. Each box represents the interquartile range (25th to 75th percentiles), the horizontal line inside is the median value, and the whiskers represent the range; (**c**) A representative Western blot is shown. Two-way ANOVA was performed. * *p* ≤ 0.05, ** *p* ≤ 0.01, *** *p* ≤ 0.001.

**Figure 3 ijms-23-05133-f003:**
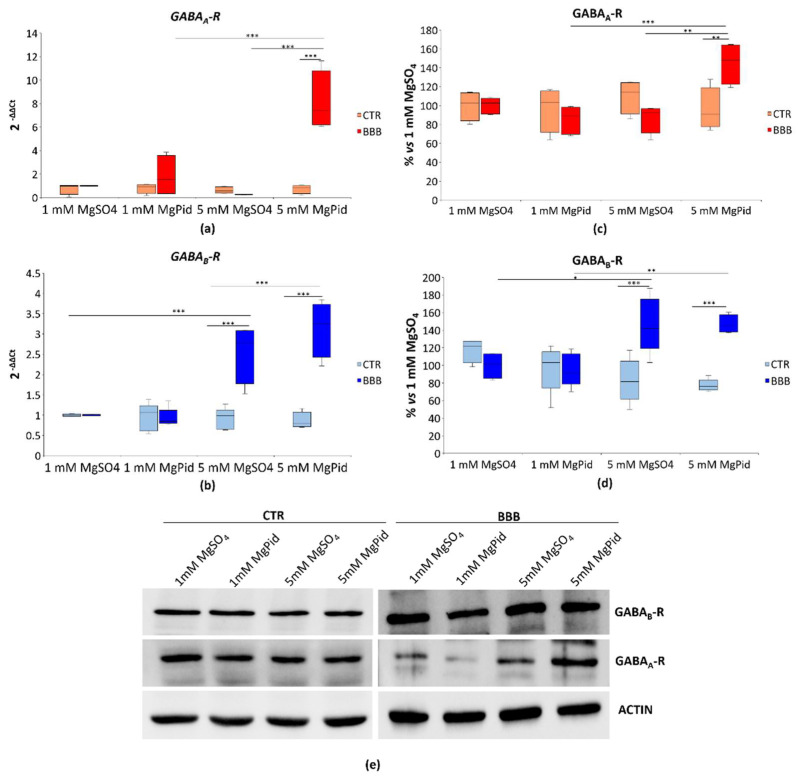
GABA-Rs in brain organoids in the presence or absence of the BBB. Human brain organoids were cultured with (BBB) or without the BBB (CTR) and exposed to 1 or 5 mM MgPid or MgSO_4_ for 4 days. Each experimental sample was obtained by pooling three organoids. (**a**,**b**) Box plot shows the results of the real-time PCR performed at least three times in triplicate on RNA extracted from the brain organoids. Primers were designed on *GABRA2* (*GABA_A_-R*) and *GABBR1* (*GABA_B_-R*) sequences, respectively. The mRNA expression values were normalized to *GAPDH* mRNA levels; (**c**,**d**) Western blot was performed using antibodies against GABA_A_-R and GABA_B_-R. Actin was used as a control of loading. The box plot shows the quantification of GABA_A_-R and GABA_B_-R of five independent Western blots. Each box represents the interquartile range (25th to 75th percentiles), the horizontal line inside is the median value, and the whiskers represent the range; (**e**) A representative Western blot is shown. Two-way ANOVA was performed. * *p* ≤ 0.05, ** *p* ≤ 0.01, *** *p* ≤ 0.001.

**Figure 4 ijms-23-05133-f004:**
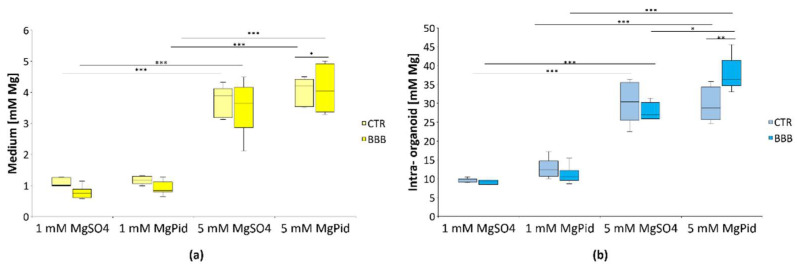
Magnesium concentration in the lower chamber and in the brain organoids in the presence or absence of the BBB: (**a**) Human brain organoids were cultured with (BBB) or without (CTR) the BBB and treated with the Mg salts for 4 days. The culture medium of the lower chamber was collected and Mg concentration was measured; (**b**) The organoids were lysed through repeated freeze–thawing cycles and 30 µg of each sample was used to measure Mg concentration. Box plot shows the quantification of at least three independent experiments in triplicates. Each box represents the interquartile range (25th to 75th percentiles), the horizontal line inside is the median value, and the whiskers represent the range. Two-way ANOVA was performed. * *p* ≤ 0.05, ** *p* ≤ 0.01, *** *p* ≤ 0.001.

**Figure 5 ijms-23-05133-f005:**
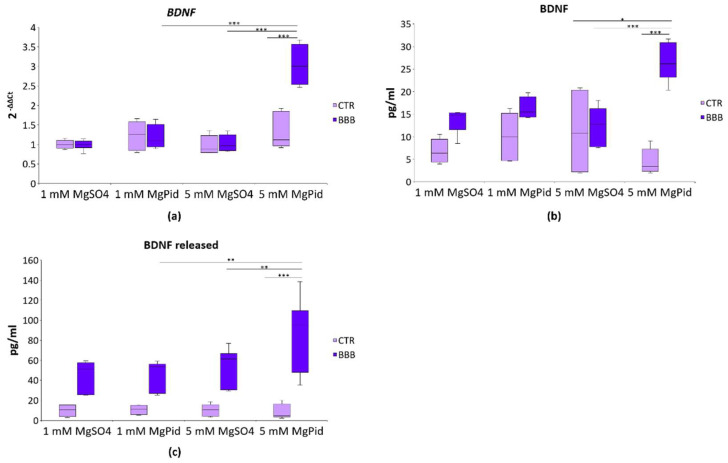
BDNF modulation in brain organoids and in the culture media in the presence or absence of the BBB. Human brain organoids were cultured in the presence or absence of the BBB, treated with 1 or 5 mM MgSO_4_ or MgPid for 4 days. Each experimental sample was obtained by pooling three organoids. (**a**) *BDNF* expression was measured using real-time PCR on RNA extracted from the brain organoids. The mRNA expression values were normalized to *GAPDH* mRNA levels. Box plot for real-time PCR data is shown; (**b**) Box plot shows the BDNF levels in the human brain organoids, as measured by ELISA; (**c**) Box plot shows the amount of released BDNF in the culture media, as measured by ELISA. Each box represents the interquartile range (25th to 75th percentiles), the horizontal line inside is the median value, and the whiskers represent the range. All the experiments were performed at least three times in triplicate. Two-way ANOVA was performed. * *p* ≤ 0.05, ** *p* ≤ 0.01, *** *p* ≤ 0.001.

## Data Availability

Data are available in a publicly accessible repository. The data presented in this study are openly available in Dataverse through the following link: https://dataverse.unimi.it/dataverse/Mg_brainorganoids/, accessed on 29 March 2022.

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
