# Peer review of "The Presence of Blood–Brain Barrier Modulates the Response to Magnesium Salts in Human Brain Organoids"

_ijms, 2022, doi:10.3390/ijms23095133_

Round 1

Reviewer 1 Report

Report on The Presence of BBB Modulates the Responses to Magnesium Salts in Human Cerebral Organoids, by Cazzaniga et al.,

This study addresses the impact of magnesium salts (in an organic and an inorganic form) on the expression of GABA-A, GABA-B and NMDA receptors, and BDNF levels in iPSC-derived brain organoids with or without a blood brain barrier. Using qPCR, Western Blot and Elisa the authors report that magnesium pidolate (5mM) increased GABA-A and GABA-B receptor expression and BDNF levels and reduced NMDA receptor expression when blood brain barrier cells (hbmec) were included in the culture model. There are grammatical errors throughout the manuscript. Moreover, it is not clear to me what hypothesis this study addresses. The physiological/pharmacological effects of magnesium ions on NMDA receptors are well established; the authors should clarify what direct effects, if any, it has on GABA receptors; they only cite a review on the role of magnesium in neurological disorders related to this rationale. Overall, the study is vague, appears to bind together an in vitro stem cell model of cerebral organoids, a blood brain barrier, magnesium salts, and disparate observations of uncertain significance; it lacks a clear hypothesis. More specific comments are below:

Title: BBB should be Blood Brain Barrier and not be abbreviated.

Introduction: The first paragraph is verbiage and should be deleted.

Line 82 and throughout the manuscript: “in the presence or not of…“ should be ‘in the presence or absence of ...’

Methods:

Line 277-278: What exactly is the human episomal Induced Pluripotent Stem Cell line purchased from Gibco? I could not find any such cell line.

Line 346-247: exactly which GABA-B and GABA-A receptor subunit antibodies were used?

Results:

Line 94: the authors say they developed a brain organoid model but give a reference to a paper by another groups’ work? Please clarify.

Figure 1: the authors should give a key on the meaning of hbmec, HA and w/o VitA; the photomicrographs are so small, they provide no useful information to the reader. Enlarge them or remove them.

Line 105: the neuroscientists who first described the action of Mg2+ on the NMDA gated ion channel are Mayer, M.L.; Westbrook, G.L.; Guthrie, P.B. (Voltage-Dependent Block by Mg2+ of NMDA Responses in Spinal Cord Neurones. Nature 1984, 309, 261–263). This is the appropriate reference.

Line 108: “In the absence of BBB ..” is meaningless. Please state what was compared – the transcript the protein or both? “On the contrary” should be ‘In contrast..’. Presumably “control” in the figure means the absence of the BBB? This should be clarified.

All figures: The lines and asterisks above the box plots should be explained in the legends; exactly what the boxes and lines indicate (medians and interquartile ranges) should be stated in the figure legends.

Line 116: why did the authors choose to look at the NMDA receptor expression after 4 days?

Line 127: why did the authors choose to look at the GABA receptor expression after 4 days?

Line146-150: The authors did not measure how much magnesium crossed into the lower chamber because they only saw differences in NMDA and GABA receptors expression when they had the BBB cells. It would seem helpful to determine how much magnesium permeates in the absence of the BBB cells in this model and eliminate a reason why they there is no changes in receptor expression (e.g. concentration of magnesium?).

Line 180: “contrasts neuroinflammation”: What does this mean?

Line 184: “not univocal”. I suspect this should be unequivocal.

Discussion: is vague and unfocussed in part because this study does not address a clearly stated hypothesis; figure 6 is unnecessary.

Author Response

We thank the reviewers for their insightful comments that prompted us to work on the manuscript. We feel that in the revised version our experimental hypothesis is highlighted and our work is presented more clearly.

REVIEWER 1

This study addresses the impact of magnesium salts (in an organic and an inorganic form) on the expression of GABA-A, GABA-B and NMDA receptors, and BDNF levels in iPSC-derived brain organoids with or without a blood brain barrier. Using qPCR, Western Blot and Elisa the authors report that magnesium pidolate (5mM) increased GABA-A and GABA-B receptor expression and BDNF levels and reduced NMDA receptor expression when blood brain barrier cells (hbmec) were included in the culture model. There are grammatical errors throughout the manuscript. Moreover, it is not clear to me what hypothesis this study addresses. The physiological/pharmacological effects of magnesium ions on NMDA receptors are well established; the authors should clarify what direct effects, if any, it has on GABA receptors; they only cite a review on the role of magnesium in neurological disorders related to this rationale. Overall, the study is vague, appears to bind together an in vitro stem cell model of cerebral organoids, a blood brain barrier, magnesium salts, and disparate observations of uncertain significance; it lacks a clear hypothesis.

We have revised the manuscript and summarized why we planned and performed these experiments. We reasoned that brain organoids co-cultured with BBB might represent an interesting experimental model to test the effects of Mg salts in the nervous system.  We based our experimental plan on some evidence deriving from clinical studies. Why analyzing the expression of NMDA-R, GABA-Rs and BDNF?

  • NMDA-R: clinicians utilize magnesium to prevent cerebral palsy (Rouse DJ, Gibbins KJ. Magnesium sulfate for cerebral palsy prevention. Semin Perinatol. 2013 Dec;37(6):414-6. doi: 10.1053/j.semperi.2013.06.025) and magnesium seems to improve functional neurological outcomes in patients with cerebral ischemia following cardiac arrest (JBI Database System. Rev. Implement. Rep. 15, 86).  Most of these effects are related to the inhibitory action of magnesium on NMDA-R activity, no data are obviously available on its expression.
  • BDNF: BDNF is important to protect the brain from injuries, and the increase of brain magnesium level is associated with the enhancement of BDNF [28].
  • GABARs:  GABAA-R, the benzodiazepine receptor, is involved in the anxiolytic-like effects of magnesium (Poleszak E. Benzodiazepine/GABA(A) receptors are involved in magnesium-induced anxiolytic-like behavior in mice. Pharmacol Rep. 2008 60(4):483-9.) and magnesium potentiates the function of GABAA-R [7].

More specific comments are below:

Title: BBB should be Blood Brain Barrier and not be abbreviated.

We have unravelled the acronym.

Introduction: The first paragraph is verbiage and should be deleted.

We have deleted the paragraph.

Line 82 and throughout the manuscript: “in the presence or not of…“ should be ‘in the presence or absence of ...’

We have corrected the manuscript.

Methods:

Line 277-278: What exactly is the human episomal Induced Pluripotent Stem Cell line purchased from Gibco? I could not find any such cell line.

We include the data sheet provided with the cells.

Line 346-247: exactly which GABA-B and GABA-A receptor subunit antibodies were used?

We have added the information about the antibodies.

Results:

Line 94: the authors say they developed a brain organoid model but give a reference to a paper by another groups’ work? Please clarify.

We have developed a model of co-culture organoid/BBB. Organoids were generated following the protocol described by Lancaster [25]. We have reworded the sentence.

Figure 1: the authors should give a key on the meaning of hbmec, HA and w/o VitA; the photomicrographs are so small, they provide no useful information to the reader. Enlarge them or remove them.

We have modified Fig 1 on the basis of your suggestion.

Line 105: the neuroscientists who first described the action of Mg2+ on the NMDA gated ion channel are Mayer, M.L.; Westbrook, G.L.; Guthrie, P.B. (Voltage-Dependent Block by Mg2+ of NMDA Responses in Spinal Cord Neurones. Nature 1984, 309, 261–263). This is the appropriate reference.

We apologize for our mistake. The reference has been corrected [8].

Line 108: “In the absence of BBB ..” is meaningless. Please state what was compared – the transcript the protein or both? “On the contrary” should be ‘In contrast..’. Presumably “control” in the figure means the absence of the BBB? This should be clarified.

We have rephrased the sentence as follows “We measured the transcript by Real Time PCR and the total protein amount by western blot in organoids co-cultured with the in vitro generated BBB (BBB) or in their controls in the absence of BBB (CTR)”.

All figures: The lines and asterisks above the box plots should be explained in the legends; exactly what the boxes and lines indicate (medians and interquartile ranges) should be stated in the figure legends.

The box plot is useful to characterize the distribution of the data. When the distribution is symmetric the median is in the centre of the box, while when it is asymmetric the median is in the upper or lower side of the box. The box represents data between the first and third quartile, while the line inside represents the median. The two segments that start at the box and extend upward and downward indicate the dispersion of values below the first quartile and above the third quartile that are not classified as outliers. The lines and asterisks indicate the statistical significance between the samples obtained using the two-way ANOVA. We add these details in the legends.

Line 116: why did the authors choose to look at the NMDA receptor expression after 4 days?

Line 127: why did the authors choose to look at the GABA receptor expression after 4 days?

The aim of our work was to study the effects of magnesium salts in organoids at the end of their maturation. Once mini-brains were mature, they were cultured under the BBB for 4 days for different reasons. i) Our previous studies showed that the BBB maintains its characteristics of high resistivity for 5-6 days. Therefore, 4 days is a reasonable time frame to work with; ii) We anticipate that exposure to high magnesium might require a time of adaptation within the organoid and in the cells composing the BBB to detect some effects; iii) Preliminary results indicate that significant release of BDNF by the BBB occurs after 48 h of culture.

Line146-150: The authors did not measure how much magnesium crossed into the lower chamber because they only saw differences in NMDA and GABA receptors expression when they had the BBB cells. It would seem helpful to determine how much magnesium permeates in the absence of the BBB cells in this model and eliminate a reason why they there is no changes in receptor expression (e.g. concentration of magnesium?).

Figure 4a of the revised manuscript shows that Mg salts accumulates in the lower chamber in the presence and absence of the BBB. We have added a new figure (Figure 4b), which shows that the intra-organoid concentrations of Mg increase after culture with 5 mM Mg salts both in the presence and in the absence of the BBB. 5 mM MgPid is slightly more efficient than MgSO4. Therefore, Mg levels alone do not explain the upregulation of GABAA-R and we propose that it is the presence of the BBB that makes the difference.

Line 180: “contrasts neuroinflammation”: What does this mean?

Magnesium is known to tame inflammatory responses, also in the brain. To make it clearer, we have changed the sentence into “Mg…prevents oxidative stress and downregulates inflammatory mediators”.

“Line 184: “not univocal”. I suspect this should be unequivocal.

We have deleted this part of the sentence.

Discussion: is vague and unfocussed in part because this study does not address a clearly stated hypothesis; figure 6 is unnecessary.

We removed fig 6. The discussion comments our finding in relation with current knowledge. In particular, we speculate about the potential implications of our results in clinical settings. The discussion also underlines that the presence of the BBB makes a difference in the response of mini brain to magnesium salts.

Reviewer 2 Report

The article by Cazzaniga et al used in vitro BBB and iPSC-derived cerebral organoids to understand the response to magnesium salts. The authors believe that the results from the pilot study is useful in understanding the neuroprotective roles of Mg. Even though the article discuss an important topic, it lacks various significant information. The article is missing many relevant experiments and will require additional experiments to make the conclusions reported. Editing the article and including the following information will increase the quality and  readability of the article.

  1. In the methods section, please describe in detail about the brain organoid protocol. ( media components used). If author has already published these information previously, please cite the article.
  2. How did author confirm the differentiation? Did author stain for any cortical layer markers? If author has published that before please cite the article.
  3. If author can include brightfield images of the neuroepithelia and the budding outgrowth mentioned that would be beneficial for the readers.
  4. It is very unclear how author generated the HA gel at the bottom of the transwell. Please explain
  5. How long the author coated the transwell with Poly-L Lysine/Fibronectin. How was it done? Where they staying on the membrane without leaking to the lower chamber?
  6. When plated the HbMEC, did you try various cell densities and how did you decide on cell density of 60,000/cm2. How did author make sure the integrity of the BBB. (checking for leakage)
  7. It ia little confusing when author mentioned Mg+ MEM. Does author added Mg on both upper and lower chambers. It is what is understood when reading  Page 9 line 317-319. If that so, 1mM Mg was alwyas present in the brain organoids? Then what is the significance of adding 1mM MgSO4 and 1mM MgPid in the upper chamber? Please explain in detail the methods.
  8. Please add a table for the primers used.
  9. Please add the details of the media components (company) through out the article.
  10. In Figure 1a please use larger BF images so that it is helpful for the readers.
  11. In all the plots, author is missing the negative controls (no Mg). This is really important and needed to be included.
  12. Fig. 4 shows that BBB completely transferred all the Mg supplied to the lower chamber. Does that mean HbMECs let all Mg pass through or the integrity was not efficient.
  13. Did author conducted experiments with just the BBB- organoids without any Mg salts control experiments to suggest that Mg has any influence on the receptors reported.
  14. Conclusion has to include more details.

Author Response

We thank the reviewers for their insightful comments that prompted us to work on the manuscript. We feel that in the revised version our experimental hypothesis is highlighted and our work is presented more clearly.
REVIEWER 2
The article by Cazzaniga et al used in vitro BBB and iPSC-derived cerebral organoids to understand the response to magnesium salts. The authors believe that the results from the pilot study is useful in understanding the neuroprotective roles of Mg. Even though the article discuss an important topic, it lacks various significant information. The article is missing many relevant experiments and will require additional experiments to make the conclusions reported. Editing the article and including the following information will increase the quality and readability of the article.
1. In the methods section, please describe in detail about the brain organoid protocol. ( media components used). If author has already published these information previously, please cite the article.
This is our first study on organoids, therefore no reference is available from our group. To generate brain organoids, we constantly refer to the protocol described by Lancaster [25] which is cited in the material and methods. We ordered the same reagents and followed step by step the protocol herein described.
2. How did author confirm the differentiation? Did author stain for any cortical layer markers? If author has published that before please cite the article.
Beyond investigating the expression of NMDA-R and GABA-Rs which are considered neuronal markers, we have characterized our organoids also by optical microscopy after staining with hematoxylin-eosin and by immunofluorescence using antibodies against S100ẞ, a marker of astrocytes, and SOX2, a marker of neuroprogenitors. These results are included in a methodological paper which is now submitted for publication. For reviewer’s knowledge we include the haematoxylin-eosin staining (A) and immunofluorescence to highlight SOX2 (B in red), the
neuronal markers GABAB-R (B, C in green) and S100β (C in red).
3. If author can include brightfield images of the neuroepithelia and the budding outgrowth mentioned that would be beneficial for the readers.
Thank you for your suggestion, we have added the images to figure 1.
4. It is very unclear how author generated the HA gel at the bottom of the transwell. Please explain
5. How long the author coated the transwell with Poly-L Lysine/Fibronectin. How was it done? Where they staying on the membrane without leaking to the lower chamber?
The set up of the in vitro generated BBB was previously described by Romeo et al. [4]. Initially, the upper side of the transwell insert is coated with 500 μL of fibronectin for 1h at 37°C. Then, fibronectin is removed, the transwell is positioned upside-down on a petri dish and the underside is coated with 800 μL of Poly-L Lysine for 1h at room temperature). Poly-L Lysine is removed and the insert is washed with water twice before seeding HA (35,000/cm2). After 3h, when HA are adherent, the transwell is inserted/overturned in the six-well plate, with HA culture medium in the lower chamber. Three days later, HbMEC are seeded on the previously fibronectin-coated
upper-side of the transwell system (60,000/cm2) in their culture medium. After three days, the BBB is ready for the experiments.
No leaking of Poly-L Lysine/Fibronectin in the upper/lower chamber occurs.
6. When plated the HbMEC, did you try various cell densities and how did you decide on cell density of 60,000/cm2. How did author make sure the integrity of the BBB. (checking for leakage)
60,000 HbMEC/cm2 were seeded because this is the number of cells necessary to obtain a confluent monolayer.
We also checked the integrity of the BBB by measuring the trans-monolayer electrical resistance (TEER) of the BBB at various time points using an EndOhm (World Precision Instruments, Friedberg, Germany) and the results were in agreement with the ones of Romeo et al [4]. These results are included in another manuscript which focuses on methodological issues.
7. It ia little confusing when author mentioned Mg+ MEM. Does author added Mg on both upper and lower chambers. It is what is understood when reading Page 9 line 317-319. If that so, 1mM Mg was alwyas present in the brain organoids? Then what is the significance of adding 1mM MgSO4 and 1mM MgPid in the upper chamber? Please explain in detail the methods.
The confusion is generated by our mistake in the methods of the previous version of the manuscript where the word “free” was omitted. We utilize customized magnesium free medium in our experiments. Using this magnesium free medium, we then add MgSO4 or MgPid to reach the final concentrations reported in the paper. The methods have been corrected.
8. Please add a table for the primers used.
In the text we specified “The following probes were used: 1. targeting GRIN1 (NMDA-R) (Hs00609557_m1), coding for the ubiquitously expressed and obligatory NMDA type subunit 1; 2. GABBR1 (GABAB-R) (Hs00559488_m1), coding for GABA type B receptor subunit 1; 3. GABRA2 (GABAA-R) (Hs00168069_m1), coding for GABA type A receptor α2; brain derived neurotrophic factor (BDNF) (Hs02718934_s1). The housekeeping gene glyceraldehyde-3-phosphate dehydrogenase (GAPDH) (Hs99999905_m1) was used as an internal reference gene.”
The probes are purchased by Life Technologies, thus we have only the catalogue number. We added the table with the codes in the text.
9. Please add the details of the media components (company) through out the article.
We selected the media of Lancaster et al. [25].
Here we report a table with the main reagents that we use to compose the media of organoids differentiation. The details are reported in Lancaster et al. [25]
11330032
DMEM NUTRIENT MIX F12
thermofisher
21103049
NEUROBASAL MED
thermofisher
10828028
KNOCKOUT(TM) SR
thermofisher
17502048
N2 SUPPLEMENT
thermofisher
12587010
B-27 SUPPLEMENT W/O VIT A 10 ml
thermofisher
17504044
B 27 SUPPLEMENT 10 ml
thermofisher
13256029
bFGF 10ug
thermofisher
35050038
GLUTAMAX
thermofisher
15140122
PENICILLIN STREPTOMYCIN
thermofisher
11140050
MEM NEAA, 100X
thermofisher
H3149-10KU
HEPARIN SODIUM CELL CULTURE TESTED
sigma
SCM075
Rock Inhibitor (Y-27632), 5 MG
sigma
I9278-5ML
Insulin Solution, Human recombinant (10ml)
sigma
10. In Figure 1a please use larger BF images so that it is helpful for the readers.
Thank you for your suggestion, we have enlarged the images.
11. In all the plots, author is missing the negative controls (no Mg). This is really important and needed to be included.
Magnesium plays crucial roles in many physiological processes such as cell growth, proliferation, differentiation, energy metabolism, and death. It is not possible to maintain cells, spheroids or organoids in 0 mM Mg medium. The control is the physiological concentration of Mg, i.e. 1 mM. This is the reason why we utilize both 1 mM MgPid and MgSO4. It could be interesting to study the effects of low Mg in brain organoids, however this is not the aim of this study.
12. Fig. 4 shows that BBB completely transferred all the Mg supplied to the lower chamber. Does that mean HbMECs let all Mg pass through or the integrity was not efficient.
We set up the BBB and assessed its integrity as described above and in the previous studies [4]. We have previously demonstrated that the BBB allows the passage of Mg. At the moment it is not clear through which pathway this happens. Studies are
ongoing to understand the role of Mg transporters –such as TRPMs, SLC1A1 and MagT1 in mediating this effect.
13. Did author conducted experiments with just the BBB- organoids without any Mg salts control experiments to suggest that Mg has any influence on the receptors reported.
The culture media of brain organoids contains about 1 mM of Mg salt and all the growth factors described in Lancaster et al [25]. Thus, for the experiments, we used as a control medium the MEM + brain organoids growth factor + 1 mM MgSO4.
14. Conclusion has to include more details.
Revising the manuscript, we decided to utilize the “conclusions” to close the discussion.

Reviewer 3 Report

Lines 302-312

You have stated that the endothelial cells and astrocytes are cultured in different media. Which medium did you use when the BBB model was constructed as you cannot use two different media in this set up as they will mix due to the pores in the transwell?

Lines 313-319

This is very confusing, the whole point is to see what affect Mg has on the organoids with and without the presence of the BBB in the set up. To test this Mg salts are added to the upper chamber of the transwell. I’m confused as to why MgSO4 was added to the lower chamber also. How can you tell what affect the BBB has on Mg transport if Mg is also added to the organoid medium? Also, the timeline from adding the Mg salts to analysing the organoids should be added to the methods section and not just mentioned in the results.

Lines 341-342

Western blotting was carried out on the samples but there is no mention of equilibrating the amount of protein in each sample. This issue can be seen clearly in Figures 2c and 3e of the main document, and Figures 2 and 3 in the supplementary data. Actin is used as a control and therefore should be of equal intensity in each sample. The authors are saying that both NMDA-R and GABA-R levels are affected by the presence of the BBB but there is also a decrease in actin levels. These blots should be repeated with total protein equalised in all samples. This would give more accurate results.

Author Response

We thank the reviewers for their insightful comments that prompted us to work on the manuscript. We feel that in the revised version our experimental hypothesis is highlighted and our work is presented more clearly.
REVIEWER 3 Lines 313-319 This is very confusing, the whole point is to see what affect Mg has on the organoids with and without the presence of the BBB in the set up. To test this Mg salts are added to the upper chamber of the transwell. I’m confused as to why MgSO4 was added to the lower chamber also. How can you tell what affect the BBB has on Mg transport if Mg is also added to the organoid medium? Also, the timeline from adding the Mg salts to analysing the organoids should be added to the methods section and not just mentioned in the results. We apologize with the reviewer for our mistake in presenting the methods. We utilize customized magnesium free medium. In the previous version the word “free” was omitted and this created confusion. Using this magnesium free medium, we then add MgSO4 or MgPid to reach the final concentrations reported in the paper. Lines 302-312 You have stated that the endothelial cells and astrocytes are cultured in different media. Which medium did you use when the BBB model was constructed as you cannot use two different media in this set up as they will mix due to the pores in the transwell?
In the upper side of the transwell, we added media containing the growth factors necessary for hbmec and the different concentrations of magnesium salts, while in the lower compartment, specific supplements of organoids [25] and the physiological concentration of Mg (1 mM MgSO4) were added to MEM. To generate the BBB model and to stabilize it, HA and HbMEC were seeded in their culture media: HA medium in the lower chamber and endothelial cell medium in the upper chamber. We are aware that the media are different, but there is the BBB in between. Thus, they are not in direct contact and can mix only minimally. The presence of the different culture media does not alter the formation of the BBB and its resistivity [4]. Then, when we add the organoids, we utilize the Mg-free MEM additioned as described above. The resistivity of the BBB is maintained. Lines 341-342 Western blotting was carried out on the samples but there is no mention of equilibrating the amount of protein in each sample. This issue can be seen clearly in Figures 2c and 3e of the main document,
and Figures 2 and 3 in the supplementary data. Actin is used as a control and therefore should be of equal intensity in each sample. The authors are saying that both NMDA-R and GABA-R levels are affected by the presence of the BBB but there is also a decrease in actin levels. These blots should be repeated with total protein equalised in all samples. This would give more accurate results. Lines 341-342 refer to Real Time PCR, 30 ng of cDNA are used to perform Real Time PCR. In the material and methods lines 356-359 it is mentioned that the amount of protein is quantified with Bradford assay and 40ug of the samples were used in western blot: “The concentration of protein in samples was assessed using the Bradford assay (Sigma Aldrich). Lysates (40 μg/lane) were separated on SDS-PAGE and transferred to nitrocellulose sheets using the Trans-Blot Turbo Transfer System following the manufacturer’s instructions (Bio-Rad)”. We have changed the blots by selecting those with a more accurate normalization. Please, consider that the representative western blots are from two different gels (CTR –BBB). However, the relevant data are the ones expressed by the box plot which summarize the results obtained in many different experiments.

Round 2

Reviewer 1 Report

The authors have improved their manuscript and addressed most of my comments. There are still errors. Here are some:

Ln 64: contrasting should be counteracting (I think).

Figure 1: embryod is embryoid

Figure 3c: lines missing over asterisks.

Ln 310-312: The human episomal Induced Pluripotent Stem Cells (iPSCs) were purchased (Gibco, Thermo Fisher Scientific, Waltham, Massachusetts, USA). I looked on the Gibco site and couldn't find any information on this 'cell line'. Please clarify what cells were used to make the iPSCs?

Author Response

REVIEWER 1 Ln 64: contrasting should be counteracting (I think). Thank you for your comment, we have corrected the text. Figure 1: embryod is embryoid Thank you for your remark, we have corrected it in the figure. Figure 3c: lines missing over asterisks. Thank you, we have modified the figure. Ln 310-312: The human episomal Induced Pluripotent Stem Cells (iPSCs) were purchased (Gibco, Thermo Fisher Scientific, Waltham, Massachusetts, USA). I looked on the Gibco site and couldn't find any information on this 'cell line'. Please clarify what cells were used to make the iPSCs? You cannot find the cells on the Gibco’ site, because these cells are temporary out of production. Here you can find some pictures of the vial we purchased from Gibco in 2020 and its relative certificate of analysis

Reviewer 2 Report

The authors addressed all my comments. But making the following corrections are necessary before publication of the manuscript.

  1. Page 10 line 348 space between “above mentioned”
  2. Page 3 line 106 it is Figure 1b instead of 2b.The same with line 108 is it Figure 1b or 2b?
  3. Figure 1b, it will be better if the illustration use a drawing of organoids other than the brightfield image.
  4. Figure 2c do not represent the Figure 2b correctly. Can authors replace it with another western?
  5. Since I was not able to find any article from the authors of previous publications with cerebral organoids, I still believe it is important for the authors to include more figures of the methods (staining) in the article.

Author Response

REVIEWER 2 1. Page 10 line 348 space between “above mentioned” Thank you, we have corrected the text. 2. Page 3 line 106 it is Figure 1b instead of 2b. The same with line 108 is it Figure 1b or 2b? Thank you for your remark. We apologize for the mistake, we have corrected the text. 3. Figure 1b, it will be better if the illustration use a drawing of organoids other than the brightfield image. Thank you for your suggestion, we have modified the figure. 4. Figure 2c do not represent the Figure 2b correctly. Can authors replace it with another western? Thank you for your comment. The box plot shows the median and the interquartile group of all the performed experiments, while figure 2c is only a representative western blot. This is the reason why figure 2c does not perfectly match the box plot. 5. Since I was not able to find any article from the authors of previous publications with cerebral organoids, I still believe it is important for the authors to include more figures of the methods (staining) in the article. Thank you for your suggestion, we have added an immunofluorescence staining in figure 1.
